# LLM-Driven Lab Result Extraction from Electronic Health Records

Mehmet F Bagci[1,2]      Toan Do[3]      Samantha R. Spierling Bagsic[5]

Anna L. Ritko[6]      Truong Nguyen[1]      Brian D. Modena[4]      Yusuf Ozturk[2]

[1]University of California San Diego, ECE Dept., La Jolla, CA 92093
[2]San Diego State University, ECE Dept., San Diego, CA 92182
[3]University of California San Diego School of Medicine, Dept. of Allergy & Immunology, CA
[4]Modena Allergy + Asthma, La Jolla, CA
[5]Dept. of Research & Development, Scripps Health, CA
[6]Dept. of Knowledge Management, Scripps Health, CA

*Abstract*—Extracting structured lab results from electronic health records (EHRs) is essential for large-scale clinical research and effective patient care, yet remains challenging due to the unstructured and heterogeneous nature of EHR data. We evaluate large language models (LLMs)—including Phi4, LLaMa-3, and Qwen2.5—across zero-shot, one-shot, few-shot, and fully fine-tuned scenarios, comparing their ability to automatically extract key lab values from unstructured clinical text. Our systematic assessment examines prompt engineering strategies, adaptation methods, and the impact of quantization on inference speed and accuracy for both pre-trained and fine-tuned models.

Notably, LLaMa-3 (8B) with full fine-tuning achieved the highest accuracy (93.79%), while LLaMa-3.3 (70B) performed best in few-shot settings, reaching 89.56% accuracy. We also demonstrate that incorporating additional spirometry data improves the accuracy of asthma severity classification, increasing it from 0.72 to 0.85. These results highlight that carefully designed prompts combined with efficient fine-tuning lead to optimal extraction performance, with model size and quantization levels introducing predictable trade-offs between speed and accuracy. This work offers actionable insights for deploying LLM-driven data extraction pipelines.

*Index Terms*—Electronic Health Records, Natural Language Processing, Large Language Models, Structured Data Extraction

## I. INTRODUCTION

Electronic Health Records (EHRs) have transformed healthcare by providing comprehensive digital repositories of patient data [1]. These records include both structured data, such as laboratory results and vital signs, and unstructured narrative text, such as clinical notes and imaging reports [2]. The widespread adoption of EHR systems over the past decade [3] has enabled large-scale retrospective cohort studies and improved disease phenotyping by offering unprecedented access to patient-level information [4]. However, this abundance of data presents significant challenges, especially due to its sheer volume and inherent complexity [5].

Traditional extraction methods, particularly with unstructured data elements, such as manual chart reviews and rule-based techniques, are often labor-intensive and susceptible to human error. These conventional approaches are not only time-consuming but also limited in their ability to scale and adapt to the variability of unstructured text. Furthermore, rule-based methods tend to be error-prone when faced with the diverse language patterns and distinctive documentation styles encountered across different healthcare settings.

The field has witnessed rapid advancements in machine learning (ML) and natural language processing (NLP) leading to more sophisticated automated extraction techniques. While recent studies have demonstrated LLMs' potential for clinical data abstraction—such as the validation-loop framework for diagnostic reports [6] and prompt-engineered pathology report parsing [7]. However, these approaches often rely on cloud-based models or lack systematic evaluation of local deployment constraints. In contrast, our work advances the field in two key ways: First, we focus exclusively on open-source models (e.g., LLaMa-3, Phi4) that can run on local servers, addressing privacy and latency barriers for healthcare applications. Second, we rigorously compare fine-tuning versus few-shot strategies across quantization levels—a critical but underexplored tradeoff for deployments.

Beyond single-task applications, a comprehensive scoping review has mapped a broad range of LLM applications in EHR processing, which include named entity recognition, information extraction, text similarity, summarization, classification, dialogue systems, and even diagnosis or prediction [8]. More recent efforts in the field have focused on developing end-to-end, distantly supervised techniques that approach human-level performance by leveraging extensive unlabeled data sets [9]. These advancements point toward a future where automated tools can efficiently process vast amounts of clinical data while maintaining accuracy.

## II. RELATED WORKS

Recent studies have advanced the extraction of structured data from EHRs and clinical reports using a variety of techniques. For instance, [10] developed a deep-learning framework for medical device surveillance using EHRs, achieving

an F1 score of 97.4% and detecting six times as many complications compared to structured data alone. While demonstrating the power of neural network approaches, their work focused exclusively on device-related events rather than lab result extraction.

Rule-based methods remain relevant in resource-constrained settings, with tools like RADEX achieving a 0.94 F1 score for radiology reports [11] and EXTEND attaining 0.92-0.96 F1 for numerical data extraction [12]. However, these systems require manual pattern updates for new data types and struggle with the linguistic variability prevalent in clinical documentation (e.g., impressions and reports with non-standardized terminology).

The advent of LLMs has introduced new possibilities, with [7] reporting 89% average accuracy for pathology stage extraction using ChatGPT. Yet critical gaps persist in the literature: Prior studies predominantly use cloud-based models, neglecting the privacy and latency requirements of clinical environments; no systematic comparison exists between fine-tuning and few-shot methods for respiratory data across model sizes; and current tools rarely evaluate how extracted data improves guideline-based decision-making (e.g., asthma severity classification per American Thoracic Society (ATS) guidelines). Our work addresses these gaps by evaluating exclusively locally deployable models, quantifying the accuracy-latency tradeoffs of quantization, particularly in the context of asthma, and demonstrating direct improvements in ATS guideline-concordant classification (accuracy: 0.72→0.85, F1 Score: 0.70→0.855).

## III. METHODS AND TRAINING PIPELINE

Our approach focuses on extracting structured data from unstructured EHR text using locally deployed LLMs. We demonstrate the effectiveness of the extracted data by improving the accuracy of guideline-driven asthma classification. In our previous work [13] to be included in the study cohort, patients were required to have a sufficient range of variables—including spirometry data, key laboratory values (e.g., eosinophil counts), and other relevant blood tests, and demographic information (age and sex)—all available within a two-year observation window. Initially, strict dependence on pre-existing structured laboratory results limited the cohort to 1,112 eligible patients. Recognizing that the main limiting factor was the lack of structured spirometry data, we applied an LLM-based extraction pipeline to the unstructured clinical notes, which yielded 14,216 additional spirometry measurements across six key parameters (FEV1 Pre %, FEV1 Pre L, FVC, FEV1/FVC ratio, FEF25-75%, and FeNO). This approach substantially improved data completeness and expanded the number of eligible patients from 1,112 to 1,687, thereby enhancing the representativeness and robustness of the study cohort.

The raw dataset, initially comprising 1.3 million unstructured text entries, included extraneous materials such as educational notes and informal communications. After applying filtering rules and keyword-based heuristics, we obtained a refined subset of 12,818 documents authored by clinicians. These included both valid and void entries, allowing us to evaluate model robustness across a quality spectrum. The dataset draws from over 30 clinical specialties, reflecting the real-world diversity of EHR documentation within a large health system. In total, 30 distinct specialties contributed data, spanning primary care, subspecialty clinics, and procedural areas. The majority of records originated from three main specialties: Allergy and Immunology (41.23%), Pulmonology (37.15%), and Pediatrics (6.49%). This distribution ensures the inclusion of both adult and pediatric respiratory care, as well as general medical and admission contexts, thus supporting a robust evaluation of data extraction tools across a clinically heterogeneous population. Notably, the EHRs analyzed were primarily free-text physician notes. Key clinical measurements, such as spirometry results, were often embedded within narrative descriptions rather than listed in structured formats. This format requires LLMs to accurately extract and interpret relevant values from complex, context-rich clinical documentation.

Figure 1 illustrates the complete pipeline. Following filtering, clinical experts manually annotated the data to support supervised learning. To assess annotation consistency, we calculated the intraclass correlation coefficient (ICC2; absolute agreement), which was 0.857, indicating strong inter-annotator reliability. The exact match accuracy between annotators was 96.88%. Additionally, when the continuous annotation values were binarized into agreement/disagreement labels, Cohen's kappa coefficient was 0.978, further confirming a high level of consistency between annotators. For ambiguous cases, discrepancies were resolved through discussion or adjudication by a third expert. We evaluated zero-shot, one-shot, and few-shot prompting, as well as fine-tuning methods. All training and inference were performed locally on secure servers, resulting in a high-quality structured dataset.

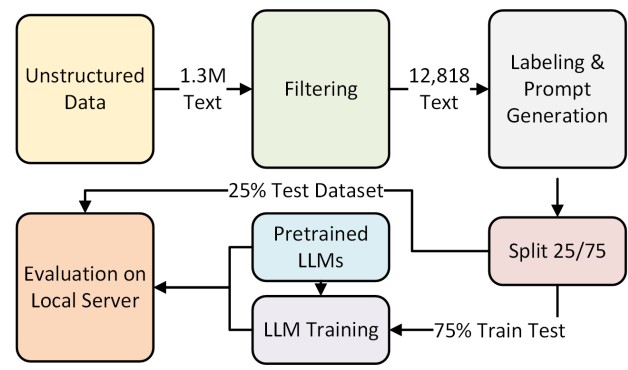

Fig. 1. Overview of the LLM evaluation pipeline, from raw EHR notes to structured output.

For prompt generation, we defined clinical targets and parsing rules in a structured JSON format to ensure deterministic outputs. Prompts were iteratively refined based on clinician feedback to improve specificity and minimize formatting errors. Prior to dataset construction, we solicited

feedback from multiple asthma specialists to refine our annotation guidelines and ensure clinical validity. We constructed three datasets—zero-shot, one-shot, and few-shot—with average input lengths of approximately 362, 674, and 1,046 tokens, respectively (measured using the LLaMA 3.1 8B tokenizer). Model training was conducted using the zero-shot format in alignment with the *TestZero* dataset.

Accurate data extraction from clinical texts relies heavily on well-structured prompts. As highlighted by [14], prompt design plays a critical role in determining the reliability and trustworthiness of LLM outputs. To maximize accuracy in extracting key clinical values, we developed a prompt format that strictly adheres to JSON standards, ensuring consistent and machine-readable responses.

Our prompting strategy includes two main components. The first is a *general task description* that outlines the extraction requirements and output format. For example:

```
{
  "task": "Extract the FEV1 percentage and
      liter values from the text",
  "rules": "Output single numeric values (
      float or string), not lists. No extra
      text or reasoning; return only JSON."
}
```

The second component involves *specific extraction rules* that guide the model in locating and interpreting individual variables. For example: *"Return the current numeric FEV1 value in liters, usually between 0 and 5 liters, sometimes without units. The value typically follows the term 'FEV1'. If there is a $-->$ symbol, use the value before it. Return it as a single float or string. If the value is not present in the text, return null."*

Model-wise, we focused on decoder-only architectures such as LLaMa [15], Phi4 [16], and DeepSeek R1 [17] due to their text-generation strengths. While encoder and encoder-decoder models were considered, they were less suited for structured output generation. For complex tasks requiring multi-step reasoning (e.g., unit disambiguation), we tested Chain-of-Thought prompting but found it inefficient for strict JSON outputs.

We evaluated locally deployable LLMs including DeepSeek R1, Phi4, Qwen2.5 [18], and various LLaMa-3 variants. Due to local hardware constraints (e.g., Nvidia A100), we prioritized models under 70B parameters. Cloud-based APIs were excluded due to privacy restrictions.

For training, we used BitsAndBytes (BNB) quantization [19] and Low-Rank Adaptation (LoRA) [20] to significantly reduce GPU memory consumption and accelerate fine-tuning. Specifically, we applied LoRA to all major transformer components, including all projection layers, ensuring maximum adaptation capacity across the model architecture. We set the LoRA rank ($r$) to 128, allowing the model to learn expressive low-rank updates, and used $lora\_alpha$ = 16 to scale update magnitude and improve training stability. A dropout rate of 0 was used to minimize regularization during adaptation.

For optimization, we primarily used the Adam optimizer [21]. However, in cases where the model exceeded memory constraints, we opted for 8-bit AdamW, which provides comparable performance while significantly reducing memory usage.

To further accelerate training, we utilized the Unsloth framework [22], which enabled memory-efficient optimization and allowed us to train with longer contexts and larger batch sizes using gradient checkpointing. Finally, we exported the trained models to the GGUF format, enabling lightweight and fast inference in C++ environments, which is ideal for deployment on resource-constrained systems.

Evaluations used Ollama [23], which supports fast GGUF-compatible inference. We compared fine-tuned outputs against prompt-only baselines using a strict numeric equality to assess accuracy and formatting consistency. We used two prompt formats: (1) conversational (user-assistant dialogue) for complex extraction tasks, and (2) Alpaca-style (single-turn instruction) for efficient batch processing. The former supported complex tasks, while the latter enabled scalable batch processing.

In adaptation, full fine-tuning updated more weights and generally achieved higher accuracy, while few-shot fine-tuning used 40 samples per task and offered efficient alternatives under constrained resources.

Lastly, our pipeline incorporated ATS guidelines for asthma severity classification. High-dose inhaled corticosteroid (ICS) use with a controller triggered further evaluation of oral corticosteroid (OCS) bursts. Two or more documented OCS bursts were used to identify severe asthma cases, following ATS guidelinesç This logic is illustrated in Figure 2. By integrating LLM-derived data with ATS guideline based patient classification accuracy increased.

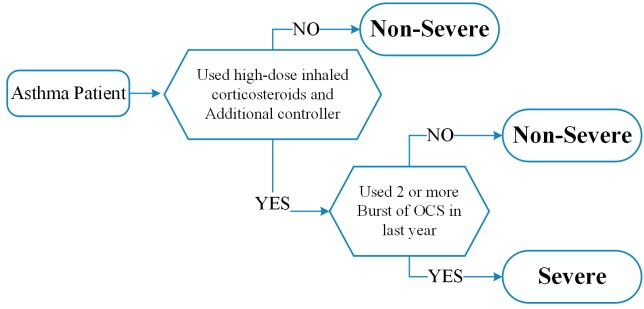

Fig. 2. Decision flow based on ATS guidelines for classifying asthma severity from structured patient data.

## IV. RESULT & DISCUSSION

The extracted data significantly enhanced our earlier study by expanding the patient cohort, which led to measurable improvements in classification model performance. Notably, the F1 score increased from 0.70 to 0.855, while overall classification accuracy improved from 0.72 to 0.85 [13]. This demonstrates the effectiveness of our refined data extraction and classification pipeline.

To clarify, all reported accuracy and F1 values in this section represent the performance of the downstream asthma severity classification model, enabled by improved extraction of key clinical features (spirometry and laboratory results) from unstructured EHR notes.

By recovering additional spirometry data from narrative notes, we expanded the eligible cohort from 1,112 to 1,687 patients, thus enhancing both the statistical power and generalizability of our analysis. Feature importance analysis revealed that several spirometry parameters—including FEF25-75% predicted, FEV1 (L), FEV1% predicted, and FEV1/FVC ratio—are among the most informative variables for asthma severity classification in our multivariate model, with FEF25-75% and FEV1 measures showing the highest relative importance.

The extracted structured data enabled guideline-based multivariate classification, and also allowed us to stratify model performance across different specialties and feature types.

These results confirm that our structured data extraction and classification approach leads to enhanced model performance, increasing both reliability and accuracy in asthma severity classification. Future work will focus on further optimizing data preprocessing techniques, implementing additional baseline comparisons, and exploring external validation strategies to generalize our findings across broader clinical datasets.

## V. ANALYSIS OF MODEL PERFORMANCES

### A. Temperature Effects

Temperature ($\tau$) is a softmax-scaling hyperparameter that controls the randomness of an LLM's output. Table I provides a clear view of how temperature settings impact model performance, as measured by extraction accuracy. Lower temperature settings (e.g., 0.1) tend to yield more deterministic outputs, which is particularly valuable for structured data extraction. The table and additional analyses reveal several key insights:

**Size-Dependent Stability**.

Larger models, such as Phi4 (14B), exhibit remarkable robustness to temperature changes, showing less than a 1% decrease in accuracy when moving from 0.1 to 0.9.

In contrast, smaller models like LLaMa 3.1 (8B) suffer more significant performance degradation, with accuracy dropping by 5.9% from 73.2% at 0.1 to 68.9% at 0.9.

**Temperature vs. Training Strategy Interaction**:

Few-shot models demonstrate a higher tolerance for increased temperatures compared to zero-shot models. For instance, Qwen2.5 (7B) experiences only a 2.0% drop in accuracy in few-shot mode, whereas LLaMa3.1 (8B) shows about an 8% decrease under similar conditions.

**Structured Output Quality**:

Lower temperature settings are critical for generating high-quality, structured outputs. They help prevent numerical value hallucinations—observed to be reduced at a temperature of 0.1—and minimize JSON formatting errors compared to 0.9.

These observations underscore the importance of carefully selecting the temperature parameter according to the specific

TABLE I
MEAN DATA EXTRACTION ACCURACY (%) ACROSS TEMPERATURE SETTINGS FOR SELECTED MODELS

| Model | Size | Temp = 0.1 | Temp = 0.5 | Temp = 0.9 |
|---|---|---|---|---|
| Qwen2.5 | 7B | 81.2 | 79.6 (-2.0%) | 78.4 (-3.5%) |
| Phi4 | 14B | 88.3 | 88.1 (-0.2%) | 87.6 (-0.8%) |
| LLaMa 3.1 | 8B | 73.2 | 71.5 (-2.3%) | 68.9 (-5.9%) |

requirements of the task. For structured extraction tasks, where reliability and exact formatting are paramount, lower temperature settings offer clear benefits by reducing randomness, curbing hallucinations, and ensuring consistent JSON formatting. In scenarios where a balance between exploration and determinism is desired, a mid-range temperature may be appropriate; however, careful calibration is essential to avoid significant drops in performance.

### B. Quantization Choices

When comparing different bit-widths, lighter quantization (e.g., 4-bit) can significantly reduce model size and speed up inference but may incur a noticeable drop in accuracy especially for pre-trained models. As shown in Table II, the degree of this trade-off varies by model. For instance, the average accuracy of the models exhibits only a minor decline when transitioning from 8-bit to 4-bit, whereas others show more pronounced performance degradation. Conversely, 16-bit often yields the highest accuracy and F1 scores, but at the cost of increased resource usage and longer call times.

Overall, 8-bit quantization (8Q) frequently emerges as an effective balance between memory footprint and predictive performance. The optimal quantization strategy ultimately depends on hardware constraints and the level of accuracy demanded by the specific use case.

TABLE II
MEAN DATA EXTRACTION ACCURACY, F1 SCORE, AND CALL TIME ACROSS QUANTIZATION LEVELS (AVERAGED OVER PHI-4, LLAMA 3.1 70B, AND LLAMA 3.3 70B)

| Quantization | Mean Accuracy (%) | Mean F1 | Call Time (s) |
|---|---|---|---|
| 4-bit (4Q) | 85.4 | 0.90 | 0.68 |
| 8-bit (8Q) | 88.2 | 0.93 | 0.79 |
| 16-bit (16Q) | 89.6 | 0.94 | 1.05 |

### C. Full Fine-Tuning vs. Few-Shot Fine-Tuning

Our experiments reveal systematic differences between full fine-tuning (Full FT) and few-shot fine-tuning (FewTRAIN) across model architectures, as shown in Table III. Overall, full fine-tuning consistently yields higher performance, with absolute accuracy gains ranging from 2.8% to 4.4% across the tested models. For example, the Phi4 (14B) model benefits the most from complete weight updates, exhibiting a 4.4% improvement in accuracy compared to its few-shot counterpart. Similarly, the corresponding increases in F1 scores (approximately 0.014 to 0.023) indicate that full fine-tuning leads to more robust and generalizable model representations.

In contrast, few-shot fine-tuning significantly reduces training costs, often achieving an 8- to 12-fold decrease in resource consumption. This efficiency comes at the expense of slightly lower accuracy and F1 scores. Moreover, full fine-tuning can enable faster inference. For instance, the Phi4 (14B) model demonstrates a marked reduction in call time, dropping from 1.42 seconds with few-shot adaptation to 0.82 seconds with full fine-tuning. This improvement in inference speed is likely attributable to more effective learned representations and the elimination of adapter overhead, which simplifies the generation process.

The interplay between model size and adaptation strategy is also noteworthy. Smaller models, such as the LLaMa 3.2 (3B), tend to exhibit a narrower performance gap between the two methods compared to larger models like Phi4 (14B). At the same time, architectures such as LLaMa 3.1 (8B) maintain relatively stable inference times regardless of the fine-tuning approach employed.

In summary, the choice between full fine-tuning and few-shot fine-tuning hinges on several factors. Full fine-tuning demands significantly more training data—typically five to ten times more—for stable convergence and higher computational resources during training. However, it offers superior performance.

TABLE III
PERFORMANCE COMPARISON OF FINE-TUNING STRATEGIES

| Model | Size | Method | Accuracy (%) | F1 |
|---|---|---|---|---|
| LLaMa 3.1 | 8B | FewTRAIN | 91.24 | 0.95 |
| | | Full FT | 93.79 | 0.96 |
| Phi4 | 14B | FewTRAIN | 88.01 | 0.93 |
| | | Full FT | 91.33 | 0.95 |
| LLaMa 3.2 | 3B | FewTRAIN | 90.21 | 0.94 |
| | | Full FT | 93.29 | 0.96 |
| Qwen 2.5 | 7B | FewTRAIN | 86.57 | 0.92 |
| | | Full FT | 91.99 | 0.95 |

*1) Fine-Tuning with 4-bit and 16-bit Models:* Our results show that well-trained models of the same architecture exhibit highly consistent performance across different quantization levels. As illustrated in Table IV, both 4-bit and 16-bit versions of LLaMa 3.1, Phi4, and Qwen 2.5 maintained nearly identical accuracy and F1 scores, with only marginal differences observed. These finding highlights that 4-bit quantization effectively preserves model performance, making it a practical and efficient choice for real-world applications.

TABLE IV
ACCURACY AND F1 SCORE ACROSS QUANTIZATION LEVELS AND
TRAINED MODELS

| Model & Quantization | Accuracy (%) | F1 |
|---|---|---|
| LLaMa 3.1 8B (4-bit) | 93.79 | 0.96 |
| LLaMa 3.1 8B (16-bit) | 93.54 | 0.96 |
| Phi4 14B (4-bit) | 92.33 | 0.95 |
| Phi4 14B (16-bit) | 93.23 | 0.95 |
| Qwen 2.5 7B (4-bit) | 91.77 | 0.95 |
| Qwen 2.5 7B (16-bit) | 91.99 | 0.95 |

### D. Comparison of Alpaca and Conversational Prompt Styles

We compared Alpaca-style [24] versus conversational prompts when fine-tuning LLaMa-3.2-3B and LLaMa-3.1-8B models. For the 3.2B model, the Alpaca prompt achieved **92.92%** accuracy, while the conversational prompt reached **93.29%**. For the 3.1-8B model, Alpaca prompts required two epochs to converge, whereas conversational prompts achieved **93.17%** accuracy in the first epoch. These results suggest that conversational prompts may offer faster convergence and improved performance.

### E. Prompting Strategies: Few-Shot, One-Shot, and Zero-Shot

Table V compares zero-shot, one-shot, and few-shot prompting strategies across various model architectures and quantization levels. Generally, few-shot prompting achieves higher accuracy and F1 scores for larger models.

TABLE V
ILLUSTRATIVE COMPARISON OF PROMPTING STRATEGIES ON MEAN
ACCURACY, F1 SCORE, AND CALL TIME

| Models | Prompt Strategy | Mean Accuracy (%) | Mean F1 | Call Time (s) |
|---|---|---|---|---|
| Phi 4 | Zero-Shot | 83.2 | 0.90 | 0.56 |
| (14B) | One-Shot | 87.2 | 0.93 | 0.66 |
| (4,8,16-bit) | Few-Shot | 88.6 | 0.93 | 0.80 |
| LLaMa 3.1 | Zero-Shot | 79.24 | 0.88 | 0.54 |
| (8B) | One-Shot | 61.6 | 0.75 | 0.45 |
| (4-bit) | Few-Shot | 61.6 | 0.75 | 0.67 |
| LLaMa 3.1 | Zero-Shot | 78.67 | 0.87 | 0.88 |
| (8B) | One-Shot | 71.47 | 0.82 | 0.66 |
| (8-bit) | Few-Shot | 68.6 | 0.79 | 0.79 |
| LLaMa 3.1 | Zero-Shot | 78.92 | 0.87 | 0.93 |
| (8B) | One-Shot | 71.36 | 0.82 | 0.58 |
| (16-bit) | Few-Shot | 68.01 | 0.79 | 0.62 |
| Qwen 2.5 | Zero-Shot | 76.81 | 0.86 | 0.54 |
| (7B) | One-Shot | 73.6 | 0.83 | 0.77 |
| (4-bit) | Few-Shot | 80.32 | 0.88 | 0.96 |
| Qwen 2.5 | Zero-Shot | 79.17 | 0.87 | 0.87 |
| (7B) | One-Shot | 80.24 | 0.88 | 0.91 |
| (16-bit) | Few-Shot | 84.35 | 0.91 | 1.06 |
| LLaMa 3.3 | Zero-Shot | 82.46 | 0.90 | 2.44 |
| (70B) | One-Shot | 85.27 | 0.91 | 2.46 |
| (16-bit) | Few-Shot | 87.82 | 0.93 | 2.58 |

However, this trend does not uniformly hold across different models. For smaller models like Qwen 2.5 (7B) and LLaMa 3.1 (8B)—particularly under 4-bit quantization—zero-shot prompting often outperforms both few-shot and one-shot methods. This suggests that compact, quantized models may face difficulties with longer or more context-heavy prompts. Conversely, larger models such as Phi-4 (14B) display stable performance with minimal variation across all prompting strategies, irrespective of quantization levels.

These results underline the dependence of prompting strategy effectiveness on model size and computational capacity. While larger models marginally benefit from additional examples, smaller models may be adversely affected, particularly in quantized, resource-efficient configurations.

### 1. Model-Scale Dependent Trends.

The effectiveness of prompting strategies is closely tied to model size. Smaller models, such as Qwen 2.5 (7B), gain significantly from few-shot prompting but often see reduced performance with one-shot prompts compared to zero-shot. Mid-scale models like LLaMa 3.1 (8B) show marked sensitivity to one-shot prompting, which can significantly degrade their performance. Larger models, exemplified by Phi-4 (14B), maintain consistent and modest improvements across zero-, one-, and few-shot prompting.

**2. Latency-Accuracy Trade-offs.**

While few-shot prompting improves accuracy, it also increases inference latency by approximately 40–70% compared to zero-shot prompting. Larger models, such as Phi-,4 scale efficiently with longer prompts, experiencing only minor latency increases (+0.24s). In contrast, smaller models like Qwen 2.5 incur disproportionately higher computational costs (+0.42s). One-shot prompting generally provides limited accuracy gains and incurs nearly equivalent latency as few-shot prompting, making it less attractive from an efficiency standpoint.

### F. Summary of Key Findings

- **Model scale dictates optimal prompting strategy:** Few-shot prompting benefits larger models but may negatively impact smaller, quantized models.
- **Quantization influences effectiveness:** Lower bit-depth (4-bit) quantization adversely impacts few-shot performance, especially in models below 10B parameters.
- **Latency-accuracy balance is critical:** Simpler prompts (e.g., zero-shot) are generally preferable for resource-constrained environments despite the potential accuracy trade-off.

Selecting the optimal prompting strategy requires careful consideration of model architecture, quantization depth, and operational constraints to balance extraction accuracy and computational efficiency.

### G. Limitations of Chain-of-Thought Models for Structured Extraction

Chain-of-Thought (CoT) models, such as DeepSeek-R1, are not well suited for structured clinical data extraction (Table VI). Their mandatory `<think>` sections disrupt strict JSON output. Furthermore, these models incur inference latencies 35–50 times higher. Overall, CoT models prioritize reasoning transparency at the cost of extraction precision, making them unsuitable for production-scale clinical NLP pipelines.

TABLE VI
PERFORMANCE OF DEEPSEEK-R1 (7B) WITH COT PROMPTING

| Strategy | Accuracy (%) | F1 | Call Time (s) | Error Rate |
|---|---|---|---|---|
| Zero-Shot | 68.92 | 0.80 | 27.39 | 31.1% |
| One-Shot | 65.34 | 0.77 | 43.11 | 34.7% |
| Few-Shot | 67.33 | 0.79 | 39.68 | 32.7% |

### H. Error Analysis

The analysis of extraction errors revealed several key challenges for LLM-based clinical information extraction, particularly for spirometry measurements. The most error-prone field was the FEV1/FVC ratio, due to two factors: when the FEV1/FVC ratio is not explicitly reported in the text, the model must infer it by dividing the FEV1 and FVC values. This type of multi-step calculation is challenging and is not feasible for traditional clinical NLP pipelines which generally require explicit value mentions. The ability to correctly perform such calculations serves as a clear differentiator between advanced and basic LLMs. The second major source of error arises from clinical texts containing multiple, often chronologically or contextually distinct, laboratory results. Models may inadvertently extract values from the wrong time point, particularly when relevant measurements are reported alongside historical or follow-up values within the same note. This issue is exacerbated when temporal cues are subtle or missing, leading to error. Errors also occur when the true value is genuinely absent from the text and the model is instructed to return null. In these situations, language models may hallucinate plausible-sounding numbers or select similar, but incorrect, laboratory results from the surrounding text. This reflects a common limitation of generative models, particularly when distinguishing between "not mentioned" and "value present but missed."

### I. Analysis of Regex Baseline Performance

The regular expression (regex) extraction method achieved a total accuracy of 79.92% on the test set. While regex rules performed with high accuracy on the training data, they showed notable limitations when applied to the test set, particularly in the extraction of the FEV1/FVC ratio. This reduction in performance can be attributed to the high variability and narrative complexity of real-world EHR text, where spirometry values may appear in diverse formats and contexts not seen during training. In particular, regex struggled when the FEV1/FVC ratio was not explicitly stated and required calculation from other measurements. These findings underscore the challenge of using rule-based approaches in unstructured clinical text and highlight the advantage of large language models for robust data extraction across varied documentation styles.

### J. Best-Performing Models Before Training

From the results, certain medium-to-large models (for example, **Phi4 14B** and **LLaMa 3.3 70B**) exhibit relatively strong out-of-the-box performance compared to smaller models like **Qwen 2.5 7B** or **LLaMa 3.2 3B**. The best model without training is the LLama 3.3 70B with a few-shot and 0.1 temperature 89.66%, F1:0.94). In particular, **Phi4 14B** with afew-shot prompts often maintains high baseline accuracy and F1 scores (88.63%, 0.934), slightly exceeding smaller LLaMa or Qwen variants before any fine-tuning. This suggests that, at larger parameter scales, the pre-training distribution captures more generalizable language patterns and better domain-relevant

representations, offering a solid starting point for further fine-tuning or prompt-based adaptation.

### K. Best-Performing Models After Training

After full fine-tuning, the LLaMa 3.1 8B model achieved the highest accuracy and F1 score (93.79% and 0.96), outperforming all other models. While the non-trained Phi4 14B model exhibited strong initial performance, additional fine-tuning further improved both Phi4 and LLaMa results. Notably, LLaMa 3.1 8B exceeded Phi4 14B and all smaller models after full training, underscoring the importance of domain adaptation. However, few-shot training alone provided only modest gains across all models, suggesting that full fine-tuning is critical for maximizing extraction accuracy from clinical text.

## VI. DISCLOSURE STATEMENT

Funding for this study was provided by GSK [NCT06389058, Study ID 219224]. Company was provided the opportunity to review a preliminary version of this publication for factual accuracy, but the authors are solely responsible for the final content and interpretation.

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
