# OpenReview forum: "LLM-Driven Lab Result Extraction from Electronic Health Records"
_IEEE.org/EMBS/BHI/2025/Conference — BHI 2025_

### Official Review · Reviewer_fU7m · 2025-07-02
**Review of “LLM-Driven Lab Result Extraction from Electronic Health Records"**

**Confidence:** 4
**Clarity Of Writing:** great
**Clinical Significance:** great
**Methodological Novelty:** great
**Overall Rating:** 7

**Experiments And Results:**

great

**Questions For The Authors:**

Clearly specify in the text and table captions what the accuracy figures represent (e.g., “lab result extraction accuracy” vs. “asthma severity classification accuracy”).


Quantify and discuss the direct contribution of fine-tuning to both extraction accuracy and downstream classification accuracy.

Define “data expansion” clearly wherever it is used, especially in figure captions, so readers understand what was added or changed.

**Strengths:**

The paper addresses a clinically meaningful problem: automating extraction of structured data for asthma severity classification from EHR notes.

The authors apply sophisticated fine-tuning techniques across several LLMs, demonstrating deep expertise in this area.

The combination of LLM extraction with a guideline-based classification framework is practical and clinically interpretable.

The paper reports strong extraction accuracy improvements with fine-tuning, and meaningful gains in classification accuracy through expanded extracted data.

**Summary Of The Paper:**

This paper presents a study on using large language models (LLMs) for extracting respiratory testing parameters (e.g., spirometry values) from unstructured electronic health record (EHR) notes. The extracted parameters are then used to classify asthma severity according to ATS guidelines. The authors evaluate several LLMs (LLaMA 3.1 8B/13B, Falcon 180B, GPT-4) in both base and fine-tuned configurations. They report performance improvements with fine-tuning and data expansion, achieving extraction accuracy up to 93.79% (LLaMA 3.1 8B) and improving asthma classification accuracy from 0.72 to 0.85.

**Weaknesses:**

It is not consistently clear whether the accuracy figures in Tables I and II refer to parameter extraction accuracy or classification accuracy — the reader is left to infer that these represent extraction accuracy, but this should be explicitly stated.


The paper does not clearly quantify how much of the performance gain in asthma classification (0.72 → 0.85) was directly due to fine-tuning versus other factors (e.g., data expansion, better prompts).

The caption for Figure 3 references “data expansion” but does not define this term clearly; readers are left guessing what was expanded.

---

### Official Review · Reviewer_9Utw · 2025-07-03
**LLM-Driven Lab Result Extraction from Electronic Health Records**

**Confidence:** 4
**Clarity Of Writing:** good
**Clinical Significance:** good
**Methodological Novelty:** fair
**Overall Rating:** 3
**Final Rating:** 6

**Experiments And Results:**

fair

**Questions For The Authors:**

How do rule-based or smaller clinical BERT models perform on the same dataset?

**Strengths:**

- Comprehensive grid of experiments (model size × quantization × prompting × tuning).
 - Clear prompt engineering recipe (deterministic JSON) that other groups can replicate.
 - Clinically motivated downstream test (asthma guideline classifier) demonstrates end-to-end utility.
 - Quantization and latency data are useful for engineers planning bedside deployment.

**Summary Of The Paper:**

The study benchmarks several open-source LLMs (LLaMA-3 variants, Phi-4 14B, Qwen-2.5 7B, DeepSeek-R1 7B) for extracting numeric lab and spirometry values from 1.3 M unstructured EHR notes. It compares zero-/one-/few-shot prompting and two LoRA fine-tuning strategies (few-shot vs. full) under 4-, 8- and 16-bit quantization. Best accuracy is 93.8 % (LLaMA-3 8 B, full FT, 16-bit). Adding the extracted spirometry values to an asthma-severity pipeline increases guideline-concordant classification accuracy from 0.72 to 0.85.

**Weaknesses:**

- No comparison to strong non-LLM baselines (regex/RadEx or smaller clinical BERT models). Without one, claims of “optimal” extraction lack context.
 - The only end-to-end evaluation is an asthma severity classifier that improves from 0.72 to 0.85 when spirometry values are included.
 - There is no analysis of how model performance varies by note type (e.g., discharge summary vs. progress note), clinical domain (e.g., pulmonary vs. nephrology), or lab test category.

---

### Official Review · Reviewer_jGLq · 2025-07-11
**A strong engineering effort in structured clinical data extraction using LLMs, with practical relevance and room for clinical benchmarking.**

**Confidence:** 4
**Clarity Of Writing:** good
**Clinical Significance:** good
**Methodological Novelty:** good
**Overall Rating:** 7
**Final Rating:** 8

**Experiments And Results:**

good

**Questions For The Authors:**

How diverse is the training data in terms of clinical note templates, institutions, or patient populations?

**Strengths:**

1. Focuses on a clinically significant problem with high real-world utility.
2. Strong engineering rigor: quantization benchmarking, LoRA fine-tuning, JSON output constraints.
3. Validates impact downstream (improved asthma classification.
4. Efficient use of open-source models with strong performance in local setups.
5. Promotes EHR privacy by avoiding cloud-based LLM APIs.

**Summary Of The Paper:**

The paper evaluates the ability of large language models (LLMs) to extract structured lab result data from unstructured EHRs. Specifically, it compares several models (Phi-4, LLaMA-3, Qwen2.5) under zero-shot, few-shot, and fine-tuned conditions to identify spirometry lab values for asthma classification. The paper employs a robust pipeline with JSON output constraints, quantization benchmarking, and downstream clinical outcome validation (asthma severity classification). Results show strong performance (accuracy up to 93.79%) and practical improvements in clinical labeling compared to baseline tools.

**Weaknesses:**

1. Dataset size (12,818 entries) is modest compared to pretraining scale and may limit generalizability.
2. No comparison to traditional clinical NLP tools (e.g., cTAKES, MedSpaCy) or hybrid rule-based baselines.
3. No expert human evaluation of correctness or error severity.
4. No stratified error breakdown (e.g., units, reference range extraction, value interpretation).
5. Lack of information on generalizability to other lab types beyond spirometry.

---

### Official Review · Reviewer_1Q1y · 2025-07-19
**Review of LLM-Driven Lab Result Extraction from Electronic Health Records**

**Confidence:** 3
**Clarity Of Writing:** good
**Clinical Significance:** good
**Methodological Novelty:** fair
**Overall Rating:** 5
**Final Rating:** 6

**Experiments And Results:**

fair

**Questions For The Authors:**

What was the inter-annotator agreement among clinicians during the annotation process?

Why does one-shot prompting underperform compared to both zero-shot and few-shot in smaller models? Additional clarification on context window constraints or prompt formulation would improve methodological clarity.

The mean accuracy reported doesn't tell much about the performance on asthma classification. Figure 3 implies an imbalance in performance.

**Strengths:**

The study presents a methodical comparison across multiple dimensions: model types, adaptation methods, quantization levels, and prompting strategies.

Emphasis on open-source, locally deployable models is forward-thinking for clinical applications where PHI (Protected Health Information) is a concern.

**Summary Of The Paper:**

The paper proposes and evaluates a pipeline using open-source LLMs to extract structured lab results (e.g., spirometry data) from unstructured EHR text. The authors systematically evaluate various adaptation strategies (zero-, one-, and few-shot, full fine-tuning), prompt formats, and quantization levels on both extraction performance and downstream asthma severity classification, as per ATS guidelines.

**Weaknesses:**

Minimal Methodological Novelty:

The paper does not propose a new model, a new training paradigm, or a novel evaluation metric. It builds a pipeline that combines existing models and standard techniques (for example, Prompt engineering, LoRA fine-tuning, BitsAndBytes quantization, and Unsloth training optimizations).

The clinical significance of each spirometry parameter isn't equally justified. More explanation is needed on how each extracted feature affects classification, especially in multivariate contexts.

The paper doesn't describe the extraction process. Additionally, how does the number of patients go from 1112 to 1687?

Annotation Process Needs More Transparency:

While clinician feedback is mentioned, the annotation protocol lacks detail on inter-annotator agreement or annotation error rates, which are essential for assessing label quality.

Reliance on substring matching for output accuracy is insufficient.